# The Bighorn Habitat Assessment Tool: A Method to Quantify Conservation Value on Landscapes Impacted by Mining

Dayan J. Anderson [1,2,*] , Vernon C. Bleich [3] and Jeffrey T. Villepique [4]

1 Department of Environmental Studies, Green Mountain College, Poultney, VT 05764, USA
2 Department of Mining Engineering, Colorado School of Mines, Golden, CO 80401, USA
3 Department of Natural Resources and Environmental Science, University of Nevada Reno, Reno, NV 89557, USA; vbleich@unr.edu
4 California Department of Fish and Wildlife, Ontario, CA 91764, USA; jeff.villepique@wildlife.ca.gov
* Correspondence: djanderson@mines.edu or onyxmining@gmail.com

**Abstract:** We present a methodology to assess the conservation value of mitigation lands for desert bighorn sheep (*Ovis canadensis nelsoni*) within landscapes impacted by historic and ongoing industrial uses. The Bighorn Habitat Assessment Tool (BHAT) was developed to support the adaptive management of the Cushenbury population of bighorn sheep located on the north slope of the San Bernardino Mountains in southern California, USA. We use a novel formulation of conservation value integrating the results of resource selection function analysis and reclamation credits, reflecting the degree to which degraded habitat is enhanced to benefit wild sheep. Our method seeks to balance conservation objectives simultaneously with the economic development of a working mine landscape. Specifically, the BHAT can be used to (a) establish a habitat reserve providing maximum benefit to the unique requirements of bighorn sheep; (b) incentivize voluntary action by industry to ensure mining activities are compatible with conservation; (c) allow for the evaluation of multiple mine planning and resource management alternatives; and (d) ensure that future compensatory mitigation actions for mining activity are grounded in the best available science. Our methodology is transferrable to the management of other wild sheep populations occupying mine-influenced landscapes for which sufficient data are available to complete resource selection analyses.

**Keywords:** biodiversity offsets; mitigation hierarchy; conservation value; decision support tools; resource selection; adaptive management; mining; desert bighorn sheep; *Ovis canadensis nelsoni*

## 1. Introduction

The suite of analytical tools available to inform conservation planning is diverse. Modeling frameworks for wildlife-habitat relationships have been developed since the 1970s [1] and range from simple nominal or ordinal ranking systems to complex computational approaches and spatial landscape simulations. Further, various indices have been developed to characterize wildlife habitat [2–6]. These frameworks are straightforward and can be applied readily at a landscape-scale [7]. With the availability of high-precision telemetry incorporating GPS technology to monitor wildlife movements, resource selection functions (RSFs) are commonly employed to estimate the relative probability that a resource unit on the landscape will be used by the species of interest [8]. This probability function, typically visualized in a raster-based format within a geographic information system (GIS), has broad applicability. RSFs have been used to develop quantitative assessments of risk to the foraging behavior of northern spotted owls (*Strix occidentalis caurina*) resulting from various timber harvest scenarios [9] and to assess the cumulative impacts of active diamond mining operations, mineral exploration activities, and seasonal outfitter camps on four species of arctic wildlife [10]. The integration of RSFs in the management of bighorn sheep (*Ovis canadensis nelsoni*) include evaluating population responses to climate change [11], predicting changes in habitat quality related to the introduction or loss of surface water

sources [12,13], evaluating the impacts of recreational land-uses [14], exploring habitat selection by ewes before and after parturition [15], and monitoring habitat use by populations reintroduced to their historic ranges [16].

Outputs of habitat modelling techniques can provide decision support for the creation of nature reserve systems that withdraw lands, inland waters, or marine areas with high conservation value from future development or extractive use. The optimal design of reserve systems having the requisite characteristics to achieve conservation objectives remains a prolific area of research. Analytical techniques have been proposed to ensure such reserves are sufficiently representative of the range of biological and environmental variation found in a region [17–21]. Methods to prioritize areas for conservation that meet the biodiversity criterion of complementarity, or the degree to which an area captures unrepresented species or habitat not already protected by existing areas, have been examined extensively [22–27]. Researchers have also proposed tools to establish reserve systems with sufficient redundancy in habitat cover type or species representation [28–30]. For example, O'Hanley et al. [31] proposed a bi-level programming method to design a reserve network that balances the criterion of complementarity with that of robustness by proportionally distributing sufficient redundant coverage across a reserve network to ensure protection of all species within the reserve in the event of catastrophic loss or damage to habitat within the reserve. These, and many other design techniques, primarily have been used to prioritize targets for conservation or to evaluate effectiveness of existing reserve networks. Despite these efforts, prioritizing conservation areas within working industrial or agricultural landscapes remains a complex problem.

Resource managers often are called upon to reconcile biodiversity conservation aims with competing uses (resource extraction, national security, renewable energy, residential growth, etc.). Numerous, and often complex, methods have been proposed to address the seemingly intractable challenge of balancing competing objectives between anthropogenic uses and natural resource conservation. As examples, Hof and Joyce [32] used mixed integer programming to find optimal timber harvest configurations that simultaneously satisfied habitat requirements for three species groups: those benefiting from non-harvested timber stands (mature growth), those benefiting from recently harvested areas (new growth), and edge-dependent species that require habitat found in both old-growth and cutover areas. Gaines et al. [33] established guidelines for designing networks of marine protected areas that would simultaneously enhance conservation and improve fishery yields and profits. Aycrigg et al. [34] used dynamic landscape simulation models to assess impacts of military training across time and space on desert tortoise populations (*Gopherus agassizii*) to determine an optimal spatial and temporal pattern that would minimize impacts on tortoises and their habitat. Additionally, Copeland et al. [35] used simulated population responses to multiple land development scenarios to prioritize conservation easements within core habitat of greater sage-grouse (*Centrocercus urophasianus*). Further, the spatial conservation tool Marxan [36] has been used to (a) optimize conservation priorities in multifunctional tropical forests in Indonesia; (b) evaluate trade-offs for multiple land-uses in Australia [37]; and (c) examine the cost-effectiveness of different agro- and silvo-environmental measures in Portugal [38].

While these examples do not represent the entire suite of analytical methods available to conservation planners, development of tools that make efficient use of available knowledge, especially in the wake of ongoing global change, remains a research and management priority. Furthermore, practical evidence-based methods that address cumulative impacts of land-use change on biodiversity are needed, especially for landscapes with mosaics of public and private ownership and for which land-use decisions are made incrementally by a multitude of stakeholders at different points in time. Planners have noted that more effective conservation outcomes can be expected if mitigation frameworks reduce regulatory hurdles [39] or offer cost savings to private developers [40]. Such frameworks must be supported by comprehensive accounting of the cumulative impacts affecting a region [39]

and an *appropriate currency* [41–44] is needed to assess trade-off exchanges between project impacts and benefits of proposed mitigation [45–47].

Our purpose is to provide land managers, industrial users, conservationists, and other concerned stakeholders with a tool that could facilitate coordinated management of a species with unique habitat requirements, but for which increasing pressures from renewable energy development, or mining and other anthropogenic land uses, may result in additional habitat fragmentation or population decline. We present a novel application of RSF analysis to assign conservation value across landscapes occupied (or potentially occupied) by bighorn sheep. Our proposed formulation establishes a basis of exchange capable of supporting systematic mitigation planning, whether on private or public lands.

## 2. Materials and Methods

### 2.1. Study Area

Our study area is located on the north facing slope of the San Bernardino Mountains, San Bernardino County, California, USA (34°20′ N, 116°54′ W; Figure 1), and has been described in detail previously [7,48,49]. Historic and ongoing mining operations overlap habitat occupied by fewer than 25 desert bighorn sheep [50] that are sympatric with mule deer (*Odocoileus hemionus*), the only other native ungulate occurring in the area [51]. Predators of bighorn sheep include mountain lion (*Puma concolor*), bobcat (*Lynx rufus*), coyote (*Canis latrans*), and golden eagle (*Aquila chrysaetos*) [52].

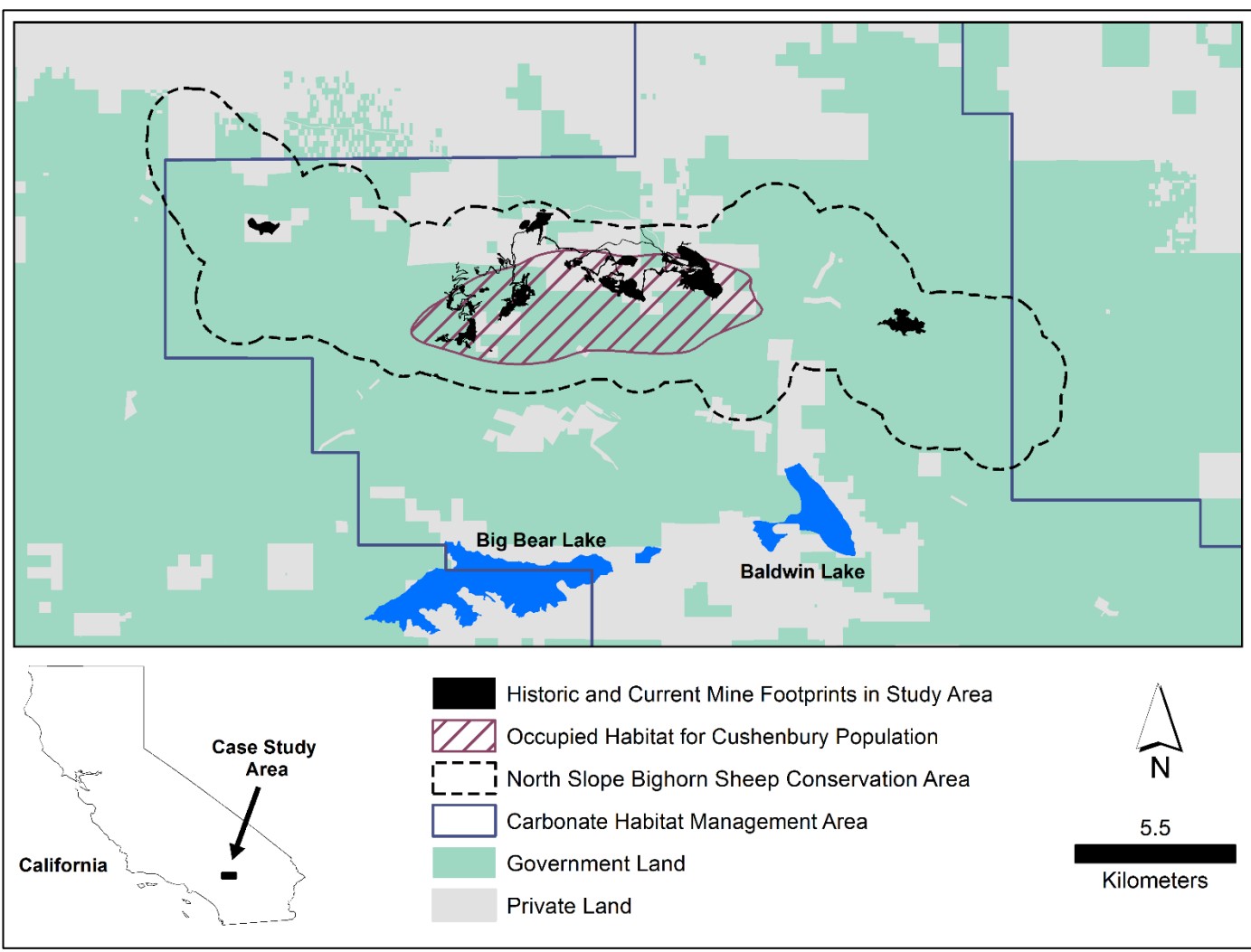

**Figure 1.** North Slope Bighorn Sheep Conservation Area, San Bernardino Mountains, California, USA.

Topographic features included desert washes, rolling foothills, steep bedrock outcrops, and talus slopes. The study area was also transected by several major canyons, and elevations ranged from 1200 m to 2500 m. Climate is characterized as Desert Transition [53]. Average annual temperature at Lucerne Valley (34°27′ N, 116°57′ W; 950 m) is 15.9 °C, with mean high and low annual temperatures of 25.7 °C and 6.2 °C, respectively [48]. On average, July ($\bar{x}$ = 27.5 °C) is the warmest month and January ($\bar{x}$ = 6.2 °C) the coolest. Average annual precipitation is 10.3 cm (range 2.5–17.8 cm); approximately 50% of annual rainfall occurs during December–February and precipitation occurs infrequently as snow during winter months. Vegetation communities transitioned from the creosote bush (*Larrea tridentata*) and blackbrush (*Coleogyne ramoissima*) series at lower elevations to conifer woodlands comprised largely of pinyon pine (*Pinus monophylla*) and juniper (*Juniperus californica*) in the upper reaches of the range [53].

Several federally listed plant taxa associated with carbonate soils, and endemic to the San Bernardino Mountains, occur within the study area [54]. As a result, resource managers engaged with active mine operations, conservation organizations, private landowners, and mining claimants to develop a strategy that balances economic interests and conservation objectives. The resulting framework, the Carbonate Habitat Management Strategy (CHMS), is a voluntary agreement to protect protected plant species while facilitating limestone mining and associated activities important to the regional economy [54].

### 2.2. Context and Purpose of the Bighorn Habitat Assessment Tool

From the baseline information developed to ensure protection of endangered plant taxa, the Carbonate Habitat Management Strategy provides for the incremental establishment of a permanent habitat reserve, with each participant in the strategy using this transparent assessment of conservation value to make decisions at different points in time [54]. No direct benefit to the Cushenbury population of desert bighorn sheep, however, is explicitly quantified. To maximize the cumulative benefit such lands would provide to these specialized plants and animals, the suitability of lands considered as mitigation for the impacts of proposed mining activity would best be evaluated in terms of their respective value to both carbonate plants and bighorn sheep.

The existing limestone mines may operate well into the next century, and precious metal prospects may bring about additional disturbance to bighorn sheep habitat if those projects advance. These cumulative pressures increase the likelihood that conservation strategies, will need to be implemented, and such efforts should be grounded in the best available science. Collaborative dialogues and research conducted over several decades [55] sought to identify and prioritize actions with the greatest potential to yield positive, effective, and long-term outcomes for the isolated Cushenbury bighorn sheep population. These efforts have culminated in the development of the North Slope Bighorn Sheep Conservation Strategy (North Slope BHSCS), an adaptive management plan developed by the California Department of Fish and Wildlife and the San Bernardino National Forest in consultation with operating mines [56]. The North Slope BHSCS is a dynamic document that governs the management of the North Slope Bighorn Sheep Conservation Area (Figure 1) and is to be reviewed every five years to facilitate inclusion and evaluation of new information as it becomes available. Mining interests have contributed to a non-wasting endowment to fund research or management actions that may be required to implement this strategy and the Bighorn Habitat Assessment Tool (BHAT) was developed to provide additional decision support to guide these efforts.

Lands proposed as compensatory mitigation for future impacts should be independently evaluated and their value quantified by the appropriate oversight agency to ensure habitat requirements for bighorn sheep and the carbonate plants are protected in perpetuity. Therefore, the BHAT is intended for use in parallel with the accounting system used to quantify conservation value to the carbonate-endemic plants, and borrows several principles and mechanisms adopted by the CHMS, among which are the use of compensation ratios based on conservation value, mitigation banking, and a recognition that disturbed

lands meeting appropriate reclamation standards can be acceptable compensatory mitigation for future projects. The formalized reserve system established by the CHMS will, hereafter, be referenced as the Carbonate Habitat Reserve to distinguish it from the Bighorn Habitat Reserve described in this paper. Considerable overlap likely will develop between these two reserves as they grow over time, and conservation should be encouraged on lands meeting the unique ecological requirements of both the endemic plant species and bighorn sheep. Further, effective reserve systems should ensure that overlapping and complementary ecological niches are represented.

Habitat analysis for the Cushenbury bighorn sheep population identified proximity to active mine areas, quarry highwalls, and revegetation sites as important determinants of selection [7,48,49], consistent with findings across North America where wild sheep and mining operations co-occur [57–66]. The BHAT recognizes that some lands disturbed and transformed by mine development can be valuable to wild sheep and, in some situations, may even be more valuable following surface perturbations and restoration [7,48,49,57–66]. For example, a mine highwall that at once provides escape terrain and high-quality or abundant forage could be of comparable, or even greater, value than an equal area of undisturbed habitat with lesser topographic relief that supports vegetation of lower nutrient value, or dense vegetation that bighorn sheep are anticipated to avoid [48,49,67]. Furthermore, researchers often have reported that bighorn sheep tolerate predictable, mining-related disturbances [48,49,57–60]. However, if this tolerance for mining activity and selection for mine features providing favorable escape terrain or improved visibility (by removal of dense vegetation) is coupled with an overall decline in forage quality or abundance, mines risk becoming ecological traps [48] if such selection results in poorer survival, nutrition, or recruitment.

Finally, some areas of historic mining within the home range of the Cushenbury population are grandfathered from any reclamation obligations because they were disturbed prior to enactment of California's Surface Mining and Reclamation Act of 1975 (SMARA, Public Resources Code, Sections 2710–2796). Those exempted sites could provide greater value to bighorn sheep than undisturbed habitat if they were enhanced with plants that increase availability of high-quality forage. Therefore, the adaptive management plan for this population should consider policy incentives that encourage improving degraded landscapes to the benefit of those specialized ungulates; the methodology presented herein provides the transparent decision support that such policy incentives require.

### 2.3. Characterization of the Landscape

With increasing use of RSF analysis to support management of bighorn sheep, the Bighorn Habitat Assessment Tool integrates RSF results into a formulation of overall conservation value for bighorn sheep occupying mine-influenced landscapes. Multiple landscape models are organized within a GIS; one reflects the current landscape, and one for each of the potential post-mining landscapes presented in the mine proposal. Each landscape is modeled as a RSF using attributes identified as being potentially important to sheep (Table A4). Formulae are applied to each unit of habitat, which is equivalent to the resolution (raster grid cell) of the RSF developed for the landscape. A *patch* of habitat is defined as a collection of grid cells being evaluated under a given management scenario, such as a proposed area of mine expansion or a land parcel to be considered for contribution to a conservation easement as mitigation for a proposed mine or activity. For each grid cell location $i$, a conservation value multiplier ($M$) is assigned using the formula:

$$M_i = B + RSF_v + R \tag{1}$$

where ($B$) is the baseline habitat value based on the cell's proximity to areas known to be used by bighorn sheep, $RSF_v$ is the probability of selection value as determined by resource selection analysis, and $R$ is a mine reclamation adjustment reflecting the extent to which disturbed lands have been enhanced for the benefit of wild sheep. The Bighorn Conservation Value ($BCV$) provided by a patch of habitat on the landscape is calculated as

the average of the grid cell values within the habitat patch, multiplied by the area (*A*) of the habitat patch:

$$BCV = A \times \left( \frac{1}{n} \sum_{i=1}^{n} M_i \right) \qquad (2)$$

where the *BCV* of a patch is expressed in bighorn conservation units (*bcu*), to be distinguished from the conservation units (*cu*) formulated by the CHMS.

Habitat on the north slope of the San Bernardino Mountains is classified in two stages, resulting in four discrete categories. Those areas within the known distribution (i.e., home range) of the Cushenbury population are defined as 'Occupied Habitat'. Areas outside of that population home range, but within the North Slope Bighorn Sheep Conservation Area (Figure 1), are classified as 'Potential Habitat' given that surrounding habitat may be enhanced for bighorn sheep as the landscape is transformed (a) by industrial activity; (b) in response to environmental factors such as climate change or wildfire; (c) as a result of management actions that influence the distribution or size of the Cushenbury population; or (d) any combination of these factors. All grid cells inside the current home range of the Cushenbury population are assigned with the maximum possible baseline habitat value (*B* = 2). Grid cells outside this home range, but within the North Slope Bighorn Sheep Conservation Area (Figure 1), are assigned a baseline habitat value of *B* = 1, while cells outside this area are assigned a baseline habitat value of *B* = 0. Lands are further classified as either 'Undisturbed Habitat' or 'Disturbed Habitat' using polygons digitized from aerial photography to delineate existing footprints of historic and current mining activity or other industrial uses. All grid cells classified as 'Disturbed Habitat' are assigned a baseline habitat value of *B* = 0.

The Bighorn Habitat Assessment Tool integrates the most recent resource selection analysis completed for the Cushenbury population, wherein each grid cell is assigned an RSF value ($RSF_V$) that is a function of fourteen different predictors of habitat selection (Table A1) expressed as a probability of use value (ranging from 0 to 1) [49]. Next, the BHAT allows for the application of mine reclamation credits, consistent with the formulations used by the CHMS in recognition that habitat impacted by mining will not always result in permanent biological losses [54]. The first three adjustments reflect the degree to which a unit of disturbed habitat has been enhanced for bighorn sheep, and the final credit relates to the timing at which those enhanced areas become available for use (Table A2). Those areas achieving the minimum revegetation standards required under existing permits are assigned an $R_M$ value of 0.25 while areas that achieve enhanced revegetation standards providing high-quality forage for bighorn sheep (Table A3) are assigned an additional increment of value ($R_E$ = 0.25). If the final quarry highwalls are designed with more wildlife access ramps than required by existing permits, they are assigned an additional 0.25-increment of value ($R_H$). The BHAT allows for incremental mitigation as the disturbance footprint of an operation increases over time. However, if a project applicant elects to submit all mitigation contributions upfront, such that new foraging areas become available to sheep sooner than would otherwise be required under existing permit obligations, an additional increment of value ($R_T$ = 0.25) is assigned to each respective contribution. The mine reclamation adjustment (*R*) applied in Equation (1) is the sum of these four increments of value, such that

$$R = R_M + R_E + R_H + R_T \qquad (3)$$

with the maximum allowable reclamation credit being *R* = 1.

## 2.4. Adjusted Conservation Value of Reserve Contributions

Fragmented reserve systems are widely recognized as being less effective than interconnected areas. In response, researchers have proposed a variety of computational methods to select land parcels in the most compact, contiguous, and spatially cohesive configuration that meet conservation objectives [68–71]. Most reserve selection algorithms work well in evaluating large datasets and usually identify solutions meeting one or more

specified constraints at a single point in time. Decisions regarding habitat conservation across landscapes characterized by mosaics of private and public ownership, however, are likely to be made by multiple stakeholder groups, whose compositions, priorities, and perspectives may be subject to change. Recognizing this, the BHAT incentivizes the creation of a continuous reserve system by assigning proposed mitigation lands that are contiguous with existing reserves a higher value than would be assigned to isolated patches of habitat and is defined here as a patch isolation adjustment (*PIA*).

Whereas the logic used by many reserve system frameworks, including the CHMS, seek to avoid fragmentation using linear, edge-based adjustments, our method looks to the social behavior of bighorn sheep as a basis for defining contiguous habitat. Provided there is no physical barrier introduced that would preclude bighorn sheep movements between habitat patches, we view reserve contributions within 200 m of any portion of an existing reserve boundary as providing sufficient spatial continuity both for social interactions within a group of bighorn sheep, and as an acceptable flight distance between those patches. The BHAT begins with the assumption that any lands within the North Slope Bighorn Sheep Conservation Area (Figure 1) currently designated for permanent conservation (even if originally set aside as mitigation for other species) are recognized as the initial Bighorn Habitat Reserve for the Cushenbury population. Legal mechanisms that allow for permanent protection might include, but are not limited to, (a) private lands whose ownership is transferred to a public land management agency; (b) private land holdings subject to a permanent conservation easement; or (c) relinquishment of a mining claim that is subsequently withdrawn from future location under the mechanisms of the CHMS. Lands considered as potential mitigation for future expansions would be evaluated in terms of their proximity to this initial Bighorn Habitat Reserve and any subsequent contributions thereto; habitat patches within 200 m of any other portion of the Reserve retain their full conservation value. However, a patch isolation adjustment (*PIA* = −0.25) is applied to all grid cells of habitat patches isolated from other recognized conservation areas at the time of the contribution.

Planning the development of a mine relies on information continually subject to revision as new geological information becomes available, market conditions change, or technological advances allow for the economic recovery of material previously classified as waste. Therefore, setting mitigation requirements that allow for some degree of operational flexibility is desired by mining interests during the permit approval process. The CHMS framework provides flexibility in this regard by recognizing two acceptable mitigation actions for the loss of carbonate plant habitat. A permanent contribution is an absolute, permanent grant of private land or relinquishment of a mining claim, and such contributions receive full credit for the conservation value those lands provide to the carbonate endemics [54]. A contributor has the option to submit a relocatable contribution, which is a temporary contribution to the Carbonate Habitat Reserve that can be replaced with other parcels of equal value in the future, but this lack of permanence incurs a cost such that the temporary contribution under the CHMS is recognized for only 50 percent of its conservation value at the time it is contributed [54]. For jurisdictional consistency, the BHAT also considers both permanent and temporary contributions to the Bighorn Habitat Reserve, referred to as a patch permanence factor (*PPF*), whereby lands not converted to a permanent conservation status undergo a reduction of 50 percent of their Bighorn Conservation Value (*BCV*).

*2.5. Determination of Compensatory Mitigation for Proposed Development*

The BHAT, as with the CHMS, applies a compensation ratio of 3:1 to quantify mitigation requirements for disturbance of habitat, such that the overall *BCV* of habitat replaced must be equal to or greater than three times the *BCV* of habitat impacted by the proposed activity. For each mine plan alternative evaluated as part of the permit application, a raster is generated to model the *BCV* of the envisioned post-mining landscape using the workflow delineated in Table A4. Topography representing the proposed post-mining

landscape for each scenario is used to update the terrain variables of each respective RSF. Next, resource managers and mine planners can model the impacts of differing reclamation plans and mitigation land configurations. Any lands within the North Slope Bighorn Sheep Conservation Area may be contributed to the Bighorn Habitat Reserve, and these lands may include either undisturbed natural habitat or disturbed lands that have been enhanced to benefit bighorn sheep. The conservation value of each potential contribution to the Bighorn Habitat Reserve is determined using the equation:

$$BCV_{Reserve\ Contribution} = A \times \left[ \left( \frac{1}{n} \sum_{i=1}^{n} M_i \right) - PIA \right] \times PPF \tag{4}$$

Temporary reserve contributions may be replaced by other lands of equal or greater conservation value. Such contributions would be a function of the reserve configuration and RSF formulation reflecting the landscape attributes and habitat use patterns of the Cushenbury population at the time of those future contributions.

## 3. Results

We developed a hypothetical landscape to demonstrate how a project proponent might satisfy its compensation requirements using the Bighorn Habitat Assessment Tool to evaluate ecological and economic alternatives. Given a proposed quarry disturbing 100 hectares within the known home range of bighorn sheep, five hypothetical mitigation sites have been identified by a mine applicant (Figure 2). Because the proposed quarry is within occupied habitat, it is assigned a baseline habitat value (*B*) of 2. The average RSF value ($RSF_V$) inside the proposed footprint is 0.73 using zonal statistics on the raster representing the existing landscape at the time of the permit application. Using Equation (2), the *BCV* inside the proposed mine footprint is equivalent to 273 bighorn conservation units (*bcu*). Using a Compensation Ratio of 3:1, the bighorn habitat mitigation requirement can be satisfied by a contribution of lands with a cumulative *BCV* of 819 *bcu*. Five hypothetical mitigation sites, each addressed individually in the following subsections, have been evaluated in terms of the benefits that the undisturbed and altered habitats, respectively provide to bighorn sheep (Table 1).

**Table 1.** Bighorn Conservation Value (BCV), expressed as bighorn conservation units (*bcu*), for five hypothetical mitigation sites on a landscape transformed by historic or ongoing mining activity. A reserve contribution formula * is applied to each site, which when aggregated, satisfies the total compensation requirement of 819 *bcu*.

| Notation | Description | Site A | Site B | Site C | Site D | Site E |
| --- | --- | --- | --- | --- | --- | --- |
| | | Extant Habitat | Extant Habitat | Rock Dump | Rock Dump | Highwall |
| | Bighorn sheep habitat status | Potential | Occupied | Potential | Occupied | Occupied |
| | Land development status | Undisturbed | Undisturbed | Disturbed | Disturbed | Disturbed |
| | Proximity to existing reserve(s) | Yes | No | Yes | No | No |
| | Permanent reserve contribution | Yes | Yes | Yes | No | No |
| *A* | Area of contribution (hectares) | 120 | 120 | 120 | 120 | 120 |
| *B* | Baseline habitat value | 1 | 2 | 0 | 0 | 0 |
| $RSF_V$ | Average RSF value of patch | 0.85 | 0.85 | 0.85 | 0.85 | 0.95 |
| $R_M$ | Meets minimum revegetation criteria | - | - | 0.25 | 0.25 | 0.25 |
| $R_E$ | Meets enhanced vegetation criteria | - | - | - | 0.25 | 0.25 |
| $R_H$ | Highwall design enhancement | - | - | - | - | 0.25 |
| $R_T$ | Accelerated mine reclamation | - | - | - | - | - |
| *M* | Conservation value multiplier | 1.85 | 2.85 | 1.10 | 1.35 | 1.70 |
| *PIA* | Patch isolation adjustment | - | −0.25 | - | −0.25 | −0.25 |
| *PPF* | Patch permanence factor | 1.0 | 1.0 | 1.0 | 0.5 | 0.5 |
| *BCV* | Bighorn Conservation Value | 222 *bcu* | 312 *bcu* | 132 *bcu* | 66 *bcu* | 87 *bcu* |

* $BCV_{Reserve\ Contribution} = A \times [(B + RSF_V + R_M + R_E + R_H + R_T) - PIA] \times PPF$.

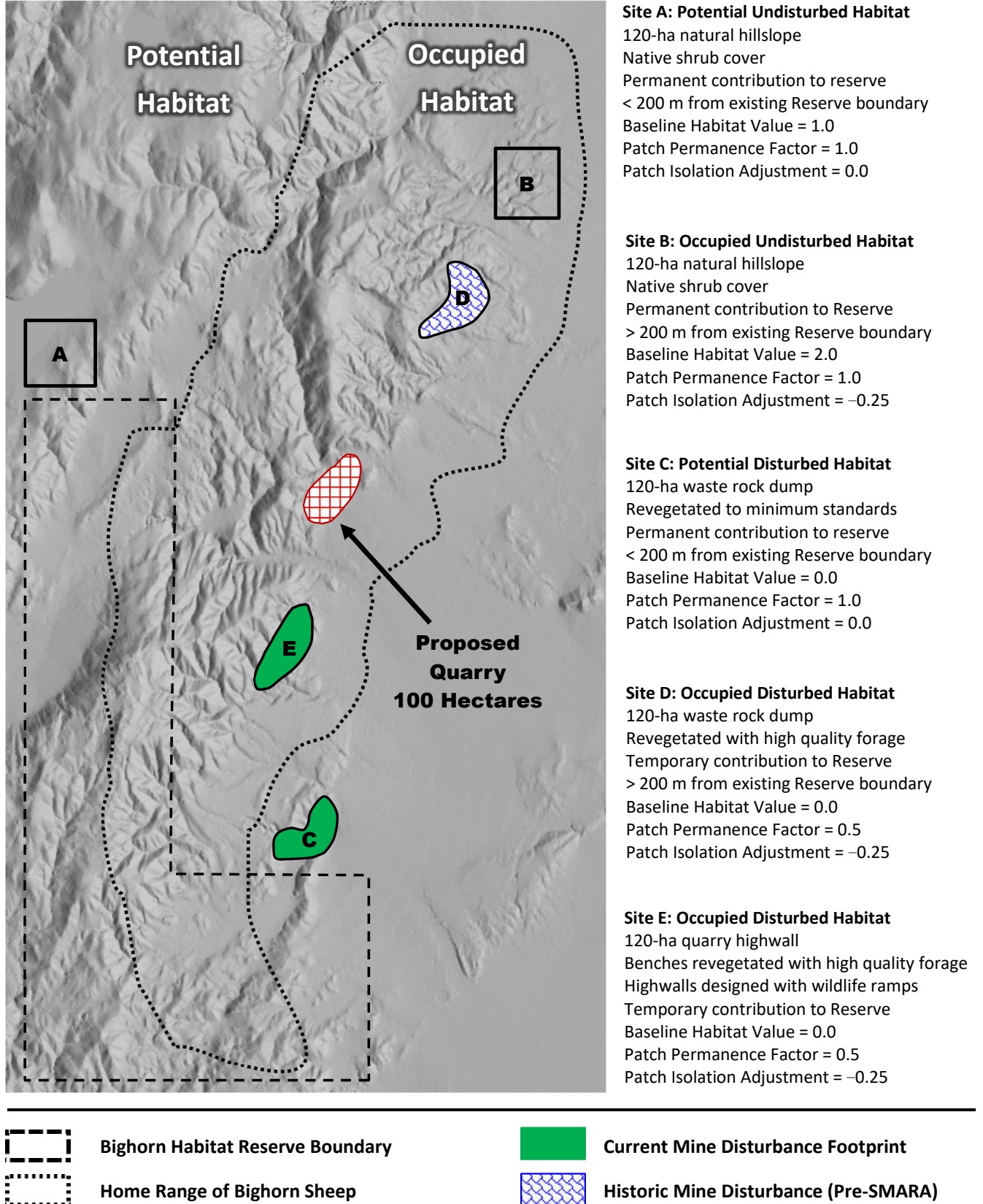

**Site A: Potential Undisturbed Habitat**
120-ha natural hillslope
Native shrub cover
Permanent contribution to reserve
< 200 m from existing Reserve boundary
Baseline Habitat Value = 1.0
Patch Permanence Factor = 1.0
Patch Isolation Adjustment = 0.0

**Site B: Occupied Undisturbed Habitat**
120-ha natural hillslope
Native shrub cover
Permanent contribution to Reserve
> 200 m from existing Reserve boundary
Baseline Habitat Value = 2.0
Patch Permanence Factor = 1.0
Patch Isolation Adjustment = −0.25

**Site C: Potential Disturbed Habitat**
120-ha waste rock dump
Revegetated to minimum standards
Permanent contribution to reserve
< 200 m from existing Reserve boundary
Baseline Habitat Value = 0.0
Patch Permanence Factor = 1.0
Patch Isolation Adjustment = 0.0

**Site D: Occupied Disturbed Habitat**
120-ha waste rock dump
Revegetated with high quality forage
Temporary contribution to Reserve
> 200 m from existing Reserve boundary
Baseline Habitat Value = 0.0
Patch Permanence Factor = 0.5
Patch Isolation Adjustment = −0.25

**Site E: Occupied Disturbed Habitat**
120-ha quarry highwall
Benches revegetated with high quality forage
Highwalls designed with wildlife ramps
Temporary contribution to Reserve
Baseline Habitat Value = 0.0
Patch Permanence Factor = 0.5
Patch Isolation Adjustment = −0.25

Bighorn Habitat Reserve Boundary

Home Range of Bighorn Sheep

Proposed Mitigation Sites

Current Mine Disturbance Footprint

Historic Mine Disturbance (Pre-SMARA)

Proposed Quarry Area

**Figure 2.** Hypothetical landscape evaluated using the Bighorn Habitat Assessment Tool (BHAT) to compare mitigation alternatives for a proposed hypothetical quarry.

### 3.1. Site Potential Undisturbed Habitat (Site A)

An area of undisturbed habitat outside the home range of the Cushenbury population is identified for contribution to the Bighorn Habitat Reserve. The 120-hectare site is located at a low elevation with rolling topography and dominated by shrubs. Although located outside the home range of the population, the site is contiguous to conservation easements previously established for the carbonate plants, which are recognized as part of the baseline Bighorn Habitat Reserve. Site A would be converted to a legal status affording it permanent protection from future development. The patch permanence factor would therefore be equal to 1 and the site would retain its full *BCV* of 222 *bcu*.

### 3.2. Occupied Undisturbed Habitat (Site B)

An area of undisturbed habitat within the home range of bighorn sheep is designated for contribution to the Bighorn Habitat Reserve. The 120-hectare site is characterized by topography with low topographical relief dominated by shrubs. The parcel is not proximate to the existing Bighorn Habitat Reserve at the time of contribution, therefore a patch isolation adjustment of 0.25 is deducted from the conservation value multiplier *M* for this contribution. Site B would be converted to a legal status affording it permanent protection from future development and thus would retain its full *BCV* of 312 *bcu*.

### 3.3. Potential Disturbed Habitat, Waste Pile (Site C)

A waste-rock dump slope located outside the home range of bighorn sheep is re-contoured with an overall slope that blends with surrounding topography. The surface of the landform is revegetated and meets the minimum performance criteria specified in existing permits and is therefore assigned a single reclamation credit ($R_M$ = 0.25). The 120-hectare site is classified as proximate (<200 m) to the existing Bighorn Habitat Reserve. The reserve contribution is permanent, and thus retains its full *BCV* of 132 *bcu*.

### 3.4. Occupied Disturbed Habitat, Waste Pile (Site D)

The waste pile for a pre-SMARA quarry within the home range of bighorn sheep is re-contoured to blend with topography and is revegetated using enhanced reclamation success criteria (Table A2) providing high-quality forage for bighorn sheep. The 120-hectare site is not proximate to the existing Bighorn Habitat Reserve and a patch isolation adjustment of 0.25 therefore is deducted from the conservation value-modifier. The private landowner voluntarily enhances the site to benefit bighorn sheep but does not wish to change the legal status of the parcel at the present time. Although the adjacent limestone is not marketable under current market conditions, or those of the foreseeable future, the owner is not ready to preclude this area permanently from future economic development. Therefore, a patch permanence factor of 0.50 is applied to this contribution, resulting in a total *BCV* of only 66 *bcu*, instead of 132 *bcu* had the contribution been permanent.

### 3.5. Occupied Disturbed Habitat, Quarry Highwall (Site E)

A highwall located on previously mined private land within the home range of bighorn sheep is proposed as a temporary contribution to the Bighorn Habitat Reserve. The horizontal surfaces of these benches are revegetated with species providing high-quality forage for bighorn sheep and achieve the enhanced species cover and richness criteria. The permittee has placed more wildlife access ramps than required by the existing permit, which results in an additional mitigation credit of 0.25. The reclamation work was completed well in advance of any new disturbance. A patch isolation adjustment (*PIA* = −0.25) is made because the highwall is not proximate to the existing Bighorn Habitat Reserve. As with Site D, the private landowner is not prepared to permanently withdraw the site from future development. Thus, the conservation value of this contribution will be reduced by 50 percent. This 120-hectare site would therefore be credited with a total *BCV* of 102 *bcu* instead of 204 *bcu* had the reserve contribution been permanent.

### 3.6. Incremental Mitigation Contributions by Mine Development Phase

Continuing with our hypothetical example, the 100-hectare quarry as proposed would advance in three sequential phases with incremental mitigation contributions in advance of each respective phase of mine development (Table 2). The first quarry phase will disturb 40 hectares of habitat, which would require a contribution of lands with a *BCV* equal to, or greater than, 328 *bcu* before that phase of development begins. The applicant could satisfy the mitigation requirements for Phase 1 with a contribution of sites A and B, resulting in a surplus of 206 *bcu* banked for the next phase of quarry development. The next 30-hectare phase of development could be mitigated by a subsequent contribution of sites C and D to meet the cumulative requirement of 573 *bcu* for the first two phases, leaving a balance of 159 *bcu* banked for Phase 3. The final reserve contribution necessary to satisfy the cumulative value of 819 *bcu* would not be required of the applicant until the third 30-hectare phase of disturbance begins. If the project applicant opted instead to accelerate mine reclamation efforts and made all required reserve contributions upfront, before any new disturbance from Phase 1 began, the conservation value multipliers for each of the mine reclamation sites C, D, and E would be assigned an additional 0.25-increment in value ($R_T$), resulting in a surplus of 60 *bcu* (Table 3). Those credits could be banked by the applicant for another expansion in the future or transferred to another party in need of additional mitigation options. The actual cost of transfer would be defined as per external business arrangements favorable to the respective parties.

**Table 2.** Five hypothetical reserve contributions used as incremental mitigation credit for three phases of a proposed quarry, expressed in bighorn conservation units (*bcu*).

| Phase of Quarry Development | Hectares Disturbed | BCV of Each Phase | BCV Mitigation per Phase | Sites Added to Reserve | Cumulative BCV Added | Banked Credits |
|---|---|---|---|---|---|---|
| Phase 1 | 40 | 109 | 328 | A, B | 534 | 206 |
| Phase 2 | 30 | 82 | 246 | C, D | 732 | 159 |
| Phase 3 | 30 | 82 | 246 | E | 819 | 0 |
| Total | 100 | 273 | 819 | | 819 | 0 |

**Table 3.** Bighorn Conservation Value (*BCV*) of reserve contributions *, expressed as bighorn conservation units (*bcu*), for five hypothetical mitigation sites on a landscape transformed by historic or ongoing mining activity. All contributions of habitat enhanced to benefit bighorn sheep are made up front before new disturbance begins, earning additional mitigation credits.

| Notation | Description | Site A | Site B | Site C | Site D | Site E |
|---|---|---|---|---|---|---|
| | | Extant Habitat | Extant Habitat | Rock Dump | Rock Dump | Highwall |
| | Bighorn sheep habitat status | Potential | Occupied | Potential | Occupied | Occupied |
| | Land development status | Undisturbed | Undisturbed | Disturbed | Disturbed | Disturbed |
| | Proximity to existing reserve(s) | Yes | No | Yes | No | No |
| | Permanent reserve contribution | Yes | Yes | Yes | No | No |
| $A$ | Area proposed for mitigation (hectares) | 120 | 120 | 120 | 120 | 120 |
| $B$ | Baseline habitat value | 1 | 2 | 0 | 0 | 0 |
| $RSF_V$ | Average RSF value of patch | 0.85 | 0.85 | 0.85 | 0.85 | 0.95 |
| $R_M$ | Meets minimum revegetation criteria | - | - | 0.25 | 0.25 | 0.25 |
| $R_E$ | Meets enhanced vegetation criteria | - | - | - | 0.25 | 0.25 |
| $R_H$ | Highwall design enhancement | - | - | - | - | 0.25 |
| $R_T$ | Accelerated mine reclamation | - | - | 0.25 | 0.25 | 0.25 |
| $M$ | Conservation value multiplier | 1.85 | 2.85 | 1.35 | 1.60 | 1.95 |
| $PIA$ | Patch isolation adjustment | - | −0.25 | - | −0.25 | −0.25 |
| $PPF$ | Patch permanence factor | 1.0 | 1.0 | 1.0 | 0.5 | 0.5 |
| $BCV$ | Bighorn Conservation Value | 222 *bcu* | 312 *bcu* | 162 *bcu* | 81 *bcu* | 102 *bcu* |

\* $BCV_{Reserve\ Contribution} = A \times [(B + RSF_V + R_M + R_E + R_H + R_T) - PIA] \times PPF$.

## 4. Discussion

Minimum performance standards to manage for *no net loss* of biodiversity have been set by financial institutions [72] and industry associations [73,74] and optimists argue that effective use of biodiversity offsets, when combined with other *additional conservation actions* [75], may achieve a *net biodiversity gain* [76]. The business community, however, remains hesitant to invest in voluntary offsets when practical challenges and technical issues related to measurement and implementation often are exacerbated by differing governmental expectations [77]. Still, for those companies intent on honoring voluntary commitments to manage toward *no net loss* or *net positive impact*, the Bighorn Habitat Assessment Tool could provide a transparent currency among regulators, permittees, and other stakeholders to use as a basis for evaluation of trade-off exchanges for economic development on landscapes used by bighorn sheep.

A means of objectively evaluating habitat suitability will support coordinated conservation planning for these specialized ungulates, especially for populations that occupy landscapes transformed by historic or ongoing resource development activity, as has occurred across the distributions of bighorn sheep [49,60,62–64,66] and thinhorn sheep (*Ovis dalli*) [65,78] in North America. Our method provides a flexible framework to aid implementation of an adaptive management plan that allows for the ongoing integration of the best available information over time, with potential application to both bighorn and thinhorn sheep. Coupled with the mitigation incentives built into the proposed formulations of a transparent currency of exchange (represented herein as the Bighorn Conservation Value, or *BCV*), an adaptive management strategy guided by the BHAT likely will encourage voluntary improvements on the existing landscape to the benefit of the Cushenbury population and other wildlife. Further, there is increasing acknowledgment that private lands play an integral role in addressing the needs of at-risk species [79–83]. The methodology recognizes both permanent and temporary contributions that private interests can make to support positive biodiversity outcomes. Moreover, with applications and assessment of appropriate variables, our method could be transferred to other large mammals dependent on landscapes currently affected by resource extraction activities [10,84–90].

As presented, the BHAT allows for incremental mitigation, providing all stakeholders the opportunity to consider added information as it becomes available. Project applicants benefit from the ability to adapt operating plans to fluctuating market conditions or new geological information, either of which may impact the timing of mining or reclamation activities. Likewise, land and wildlife managers benefit from the ability to assess the cumulative impacts of changing environmental conditions, such as those resulting from a weather or wildfire event, or consequences of climate change, as well as cumulative impacts of neighboring landscape modifications. Additionally, all stakeholders are afforded an opportunity to develop insights from biological and ecological research as results become available and have the potential to improve overall efficacy of mine reclamation activities. Considering challenges to conservation in general, and the multitude of issues facing wild sheep in particular [78,91,92], " . . . *we can no longer afford fragmentation in our management any more than we can afford habitat fragmentation in natural ecosystems*" [93]. Whether from the standpoint of habitat management or population management, proactive and cooperative efforts among government agencies, the conservation community, private landowners, industry, academic institutions, and legislators will enhance the efficacy of any efforts to conserve not only wild sheep, but all species of wildlife [80,94].

## 5. Conclusions

The methodology underlying the Bighorn Habitat Assessment Tool may be readily adapted to other mine-influenced landscapes occupied by bighorn sheep or other species for which sufficient data are available to develop a resource selection function specific to the respective landscape. Bighorn sheep, Dall's sheep, and other ungulates may function as metapopulations in some geographic settings [77,94–97]. To be effective, any biodiversity offset intended for their benefit must be valid at the larger metapopulation scale,

not just within the immediate home range of the population impacted by development. Such a framework should also allow for, and assign sufficient value to, contributions that (a) maintain movement corridors between sub-populations; (b) enhance degraded landscapes occupied by subpopulations comprising the metapopulation; and (c) support the conservation of habitat used by other metapopulations of wild sheep or the particular species of interest. With the incorporation of RSF analysis as part of the quantification of conservation value, our method accounts for selection by each subpopulation on its respective range, with the highest values assigned to landscape features and resources most strongly selected in each landscape. Further, our combined use of RSFs and reclamation credits can incentivize improvement of abandoned mine landscapes that many species are known to use [98]. By adopting frameworks that recognize voluntary, private-sector efforts to enhance landscapes to the benefit of mountain sheep or other species of large mammals, land and wildlife managers have an opportunity to influence positive long-term conservation outcomes for these specialized ungulates, as well as flora and other fauna within those landscapes.

**Author Contributions:** Conceptualization, D.J.A.; methodology, D.J.A.; writing—original draft preparation, D.J.A.; writing—review and editing, D.J.A., J.T.V. and V.C.B.; visualization, D.J.A. and J.T.V.; supervision, J.T.V. and V.C.B. All authors have read and agreed to the published version of the manuscript.

**Funding:** The development of the BHAT received no external funding. Deployment of GPS collars and generation of the underlying Resource Selection Function used in this study was funded by Mitsubishi Cement Corporation and the Big Game Management Account of the California Department of Fish and Game (now the California Department of Fish and Wildlife).

**Institutional Review Board Statement:** Not applicable.

**Informed Consent Statement:** Not applicable.

**Data Availability Statement:** No new data were created or analyzed in this study.

**Acknowledgments:** Support in the form of funding, logistics, data collection, and volunteer "boots on the ground" necessary for the ongoing management of the Cushenbury population have been provided by numerous organizations since the 1970s, among which are the Society for the Conservation of Bighorn Sheep, Sacramento Safari Club, San Bernadino Fish and Game Commission, Victor Valley College, Mitsubishi Cement Corporation, Specialty Minerals Inc., Omya California, and the California Department of Fish and Game (now the California Department of Fish and Wildlife). This is Professional Paper 140 from the Eastern Sierra Center for Applied Population Ecology.

**Conflicts of Interest:** The authors declare no conflict of interest.

## Appendix A

**Table A1.** Logistic regression coefficients estimated from resource selection function (RSF) for desert bighorn sheep (*Ovis canadensis nelsoni*) in proximity to active limestone mining operations in the San Bernardino Mountains, California, USA, 2006–2009 [49].

| | | | Odds Ratio—95% CI | | | |
|---|---|---|---|---|---|---|
| **Variable** | **Estimate** | **SE** | **Estimate** | **Lower** | **Upper** | **Bighorn Sheep Select Locations** |
| BAR | 0.4917 | 0.0354 | 1.64 | 1.53 | 1.75 | In areas with barren cover |
| CVX150 | 0.3429 | 0.0178 | 1.41 | 1.36 | 1.46 | With convex topography over a 150-m radius |
| DH2O | −0.5487 | 0.0211 | 0.58 | 0.55 | 0.60 | Closer to point water sources |
| DMFA | −0.7595 | 0.0177 | 0.47 | 0.45 | 0.48 | Closer to the active mining areas |
| DVEG | −0.5408 | 0.0224 | 0.58 | 0.56 | 0.61 | Closer to revegetated mine areas |
| ELEV | 0.1073 | 0.0046 | 1.11 | 1.10 | 1.12 | At higher elevations |
| HIWALL | 0.6018 | 0.0388 | 1.83 | 1.69 | 1.97 | On quarry highwalls |
| MINOTH | 0.1363 | 0.0348 | 1.15 | 1.07 | 1.23 | Within other mine areas |

**Table A1.** *Cont.*

| | | | Odds Ratio—95% CI | | | |
|---|---|---|---|---|---|---|
| **Variable** | **Estimate** | **SE** | **Estimate** | **Lower** | **Upper** | **Bighorn Sheep Select Locations** |
| MIX | −0.4905 | 0.0524 | 0.61 | 0.55 | 0.68 | Outside mixed conifer-hardwood cover |
| PITBOT | −1.2078 | 0.1263 | 0.30 | 0.23 | 0.38 | Outside quarry pit bottoms |
| RUG100 | 0.3358 | 0.0239 | 1.40 | 1.34 | 1.47 | With rugged topography over a 100-m radius |
| SHB | 0.3651 | 0.0211 | 1.44 | 1.38 | 1.50 | In areas of shrub vegetation |
| SLOPE | 0.0288 | 0.0008 | 1.03 | 1.03 | 1.03 | With steeper slopes |
| YRSFIRE | 0.0083 | 0.0005 | 1.01 | 1.01 | 1.01 | Without recent wildfire |

**Table A2.** Reclamation credits (R) assigned to habitat transformed by mining based on the extent to which reclamation activities benefit wild sheep.

| | **Habitat Characteristic** | **Value** | **Description** |
|---|---|---|---|
| $R_M$ | Meets minimum revegetation standards | 0.25 | Baseline credit for revegetated areas satisfying minimum success criteria specified under existing permits providing some forage value |
| $R_E$ | Meets enhanced revegetation standards | 0.25 | Additional credit for sites enhanced with species of higher forage quality and abundance than required by existing permits |
| $R_H$ | Provides more interconnected highwall access | 0.25 | Additional credit for highwalls designed with more wildlife access ramps than required by existing permits |
| $R_T$ | Provides all benefits upfront in advance of any new habitat loss | 0.25 | Additional credit for accelerated reclamation of degraded lands before new disturbance begins |

**Table A3.** Proposed performance criteria for mine revegetation sites contributed to the Bighorn Habitat Reserve.

| **Performance Criteria** | **Minimum Revegetation Threshold** [1] | **Threshold Enhanced for Bighorn Sheep** |
|---|---|---|
| Revegetation Islands [2] | Topsoil Islands will cover 30% of revegetation site | Topsoil Islands will cover 50% of revegetation site |
| Native tree and shrub cover of islands | At least 50% of pre-disturbance cover of reference areas | At least 75% of pre-disturbance cover of reference areas |
| Native Plant Species Richness | At least 50% of native tree and shrub species richness of reference area | At least 50% of the species planted will be of high-forage quality for bighorn sheep [2] |
| Hydroseeding of slopes where feasible | No definitive criterion set because sites cannot be safely monitored | At least 50% of species included in hydroseed application will be of high forage quality [3] |

1. Revegetation performance criteria vary across the three mine operations that were permitted at different points in time. Each operator would be expected to meet the criteria of their respective permits. To earn credit for voluntary enhancements, the minimum thresholds proposed would apply to sites mined prior to the 1975 Surface Mining and Reclamation Act (SMARA) that otherwise are not subject to any specified guidelines. 2. Prior to the enactment of SMARA, salvaging of topsoil was not a standard practice in the mining industry. Further, rocky outcrops characterize much of the landscape, compounding a significant deficit in availability of topsoil. Consequently, reclamation plans call for a patchwork of vegetated "islands" containing native seed to provide a source for surrounding substrates. 3. Some plant species, although not preferentially selected by sheep, may be necessary to meet the permittee's species diversity requirements and still serve important ecological functions on the successional path for the site. Therefore, upon completion of diet forage quality analyses, species selection by permittee should be carried out in consultation with the California Department of Fish and Wildlife and the US Forest Service to determine optimal species selection and/or if these criteria require modification.

**Table A4.** Proposed geoprocessing workflow of the Bighorn Habitat Assessment Tool (BHAT).

| Workflow Step | Layer Name | Description |
|---|---|---|
| 1. Create baseline habitat value raster | B.img | Create raster for the North Slope Bighorn Sheep Conservation Area; grid cells within home range (Occupied Habitat) receive baseline value of B = 2, those outside range (Potential Habitat) receive a value of B = 1. |
| 2. Reclassify disturbance areas | | Areas within current mine disturbance footprints or other notable areas of "degraded" habitat are reclassified with a baseline habitat value of B = 0. |
| 3. Extend RSF raster | RSF00.img | Create resource selection function (RSF) raster with the same extents as the Baseline Habitat (B.img) raster created in Step 1. |
| 4. Raster Math | L00.img | Add the base raster (B.img) and RSF raster (RSF00.img) to establish the reference landscape for analysis of proposed quarry and mitigation alternatives. |
| 5. Quantify Bighorn Conservation Value | | Perform Zonal Statistics on L00.img to determine "mean" conservation value within proposed disturbance footprint(s) and apply formulae to quantify Bighorn Conservation Value (BCV). |
| 6. Update terrain rasters for each mine alternative | | Use 3D mine plan provided by applicant to update terrain layers using final topography upon mine closure: elevation (TE00.img), slope (TS00.img), ruggedness (TR00.img), and convexity (TC00.img). |
| 7. Update vegetative cover | | Re-classify raster cells within the proposed project area with a "barren" cover type for inclusion in the RSF raster representing the post-mining landscape. |
| 8. Update mine cover type | | Classify proposed mine-related features using same categories of RSF (i.e., highwalls, quarry pit floors and other disturbance areas). |
| 9. Update water sources | | Create new "distance to" water layer to represent water source(s) that may be removed or added as part of mitigation plan for proposed project or other planned management actions. |
| 10. Update raster for revegetation areas | | Re-classify raster cells within the proposed project area with a "REVEG" cover type for inclusion in the RSF raster representing the post-mining landscape. |
| 11. Create post-mining raster | RSF01.img | Apply regression coefficients from most recent RSF analysis to updated terrain, vegetation and mine cover layers and generate a new post mine-closure RSF surface. |
| 12. Create reclamation raster | R01.img | Create a raster representing the mine closure plan and assign reclamation credits reflecting the extent to which each site would be enhanced to benefit bighorn sheep. |
| 13. Raster Math | L01.img | Add the base raster (B.img), post-mining RSF raster (RSF01.img) and reclamation raster (R01.img) to establish the new hypothetical landscape (L01.img) for comparative analysis against other alternatives. |
| 14. Quantify value of mitigation lands | | Perform Zonal Statistics on L01.img to determine "mean" conservation value within proposed mitigation footprints and apply formulae to quantify Bighorn Conservation Value (BCV) for that alternative. |

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
