# Peer review of "The Bighorn Habitat Assessment Tool: A Method to Quantify Conservation Value on Landscapes Impacted by Mining"

_land, doi:10.3390/land11040552_

Round 1
Reviewer 1 Report
The topic is interesting from the point of view of biodiversity, conservation planning and compensatory mitigation.
The authors contributed a new approach to the research which is based on methodology to assess the suitability of mitigation lands as bighorn sheep habitat.
Comments:
- The introduction lacks information about the global significance of the presented research results. I mean, how the international audience could benefit from the solutions presented by you.
- Figures 1 and 2 are not of good resolution. In addition, figure 2 needs to be improved, in this version the drawing is too infantile.
- The discussion definitely lacks a comparison to other studies in this topic.
Author Response
Point 1: The introduction lacks information about the global significance of the presented research results. I mean, how the international audience could benefit from the solutions presented by you.
Response 1: We appreciate the comments conveyed by Reviewer 1. In response, we have expanded the introduction to include suggestions for application on an international level, and provide additional details regarding the utility of our methodology, particularly as it relates to advantages associated both with conservation and economic issues.
Point 2: Figures 1 and 2 are not of good resolution. In addition, figure 2 needs to be improved, in this version the drawing is too infantile.
Response 2: Comment acknowledged, and figures of increased resolution have been prepared.
Point 3: The discussion definitely lacks a comparison to other studies in this topic.
Response 3: To the best of our knowledge, this methodology is the first to have been developed in the context of habitat conservation specifically for a large mammal. We have included additional information in the introduction (see our response to Point 1, above), and reemphasized the potential benefits of the application of this model in the management implications.
Reviewer 2 Report
General comments
The manuscript deals with a topic of great interest, both from a scientific point of view and from a more strictly practical point of view.
Each conceptual proposal is important for the development of an approach to conservation that takes into account specific solutions for specific species and environments.
Therefore, I believe the proposed study is an important contribution in the field and the manuscript only needs minor corrections.
In general, the paper is well structured, very well written, with extreme clarity and accuracy. But I think the recommended structure of the journal should be followed more closely: " all manuscripts must contain the required sections ………, Introduction, Materials & Methods, Results, Conclusions " (https://www.mdpi.com/journal/land/instructions).
Moreover, the paper could be enriched by inserting, perhaps as additional content, some important elements for the evaluation of the proposed methodology, which can be found in the reference bibliography for their consultation (for example the fourteen different predictors of habitat cited in L-251).
Author Response
Point 1: In general, the paper is well structured, very well written, with extreme clarity and accuracy. But I think the recommended structure of the journal should be followed more closely: " all manuscripts must contain the required sections ………, Introduction, Materials & Methods, Results, Conclusions " (https://www.mdpi.com/journal/land/instructions).
Response 1: We very much appreciate the very positive evaluation provided by Reviewer 2. In response to this suggestion, we have considered ways in which we can comply more closely with the structure of the journal, but we believe the subheadings we included under Methods help substantially with the organization of the paper. Upon review, however, we have shortened the discussion, and added a sub-heading (conclusions) in which we summarize the utility of the method described in the paper, and its potential application to other species of large mammals that may be impacted by resource extraction activities, and how application of the tool can benefit both wildlife conservation and industry.
Point 2: Moreover, the paper could be enriched by inserting, perhaps as additional content, some important elements for the evaluation of the proposed methodology, which can be found in the reference bibliography for their consultation (for example the fourteen different predictors of habitat cited in L-251).
Response 2: We have included Table A1 in the Appendix that summarizes the 14 predictor variables identified by Anderson et al. 2017 and we expanded our discussion of important habitat factors for sheep, supplemented with additional examples/citations of studies conducted on landscapes where wild sheep and mining co-exist.
Reviewer 3 Report
Dear Authors,
I think that the article has a scientific sense and the knowledge presented can be useful for the protection of mountain areas exploited by mines. The literature looks appropriate and is well presented.
Unfortunately, the article is not very well structured. First, I propose to add a case study in the title. The presented concept is shown only for one area. I also do not find an explicit purpose for the work, which should be added before the Methodology.
The Methods section should be named Materials and Methods. First describe the area of analysis and then the methods used to evaluate it. Some of the paragraphs presented by you (e.g. 3.1) are more suitable for discussion than for Research Methods.
Improve the quality of Figure 1. As it stands, it is little readable.
There is no research results and discussion section, even though you refer to studies by other authors in various parts of the article. P. 3.4; 3.4.1 are research results resulting from your assumed methodology.
Finally, the Conclusions are too long. Do not discuss with them. They should correspond to the aim of the research.
Author Response
Point 1: I think that the article has a scientific sense and the knowledge presented can be useful for the protection of mountain areas exploited by mines. The literature looks appropriate and is well presented. Unfortunately, the article is not very well structured. First, I propose to add a case study in the title. The presented concept is shown only for one area. I also do not find an explicit purpose for the work, which should be added before the Methodology
Response 1: We appreciate the helpful comments provided by Reviewer 3. In response, we have modified the title as suggested to reflect the concern that this is a case study, and we added additional text to clarify the potential value of this method both to wildlife and to industry.
Point 2: The Methods section should be named Materials and Methods. First describe the area of analysis and then the methods used to evaluate it. Some of the paragraphs presented by you (e.g. 3.1) are more suitable for discussion than for Research Methods.
Response 2: Admittedly, this paper does not entirely fit within the categories typically associated with scientific papers (Introduction, Methods, Materials, Results, Discussion, and Conclusions. As per the suggestion of Reviewer 2, we have modified the Discussion, and added a section titled Conclusions, in which we emphasize the utility of our model across taxa and habitat types. Please see, also, our responses to Point 3, which we believe rectifies this concern.
Point 3: Improve the quality of Figure 1. As it stands, it is little readable.
There is no research results and discussion section, even though you refer to studies by other authors in various parts of the article. P. 3.4; 3.4.1 are research results resulting from your assumed methodology.
Response 3: We have produced new and clear figures, as requested. Further, we have modified the organization structure and it now includes a section (Results) in which we apply the technique developed and explained in "Methods" to five situations with which a hypothetical quarry is faced. Following this is a more general Discussion, and a penultimate section (Conclusions) has been added in which we emphasize the broad utility of the Habitat Assessment Technique and its potential for application in a variety of settings.
Point 4: Finally, the Conclusions are too long. Do not discuss with them. They should correspond to the aim of the research.
Response 4: A separate section (discussion) has been added, and the overly long Conclusions have been substantially shortened, as per the reviewer's suggestion.
Round 2
Reviewer 3 Report
Dear Authors
I like the good changes you have made, which have definitely improved the scientific quality of the manuscript. However, I have 2 minor comments.
1. The purpose of the paper is still not clearly stated at the end of the introduction.
2. Pages 15; 16 and 18 are blank. There was some text formatting error.
Author Response
Thank you again for your thoughtful review. We moved some material from our Methods section to the Introduction to better set the larger context for our study, and have added a short paragraph at the end of the Introduction to explicitly state the purpose of our work.